# Identifying priority review questions for Cochrane Eyes and Vision: protocol for a priority setting exercise

Jennifer R Evans ,[1,2] Iris Gordon,[1,2] John G Lawrenson,[3] Roses Parker,[4] Fiona J Rowe ,[5] Gianni Virgili,[1,6] Tianjing Li,[7] Jacqueline Ramke ,[2,8] The Cochrane Eyes and Vision Priority Setting Steering Group

► Prepublication history and additional online supplemental material for this paper are available online. To view these files, please visit the journal online (http://dx.doi.org/10.1136/bmjopen-2020-046319).

## ABSTRACT

**Introduction** Cochrane Eyes and Vision (CEV) is an international network of individuals working to prepare, maintain and promote access to systematic reviews of interventions to treat, prevent or diagnose eye diseases or vision impairment. CEV plans to undertake a priority setting exercise to identify systematically research questions relevant to our scope, and to formally incorporate input from a wide range of stakeholders to set priorities for new and updated reviews.

**Methods and analysis** The scope of CEV is broad and our reviews include conditions that are common and have a high global disease burden, for example, cataract and dry eye disease, and conditions that are rare but have a high impact on quality of life and high individual cost such as eye cancer. We plan to focus on conditions prioritised by WHO during the development of the Package of Eye Care Interventions. These conditions were selected based on a combination of data on disease magnitude, healthcare use and expert opinion. We will identify priority review questions systematically by summarising relevant data on research in Eyes and Vision from a range of sources, and compiling a list of 10–15 potential review questions (new and/or updates) for each condition group. We will seek the views of external and internal stakeholders on this list by conducting an online survey. Equity will be a specific consideration.

**Ethics and dissemination** The study has been approved by the ethics committee of the London School of Hygiene & Tropical Medicine. We will disseminate the findings through Cochrane channels and prepare a summary of the work for publication in a peer-reviewed journal.

## Strengths and limitations of this study

► This is a systematic assessment of priority questions for Cochrane Eyes and Vision systematic reviews.
► We will seek global input, considering questions irrespective of location or setting.
► The study will draw on a wide range of stakeholders and resources.
► The focus will be mainly on new reviews and topics but we will also consider potential review updates.
► The conduct of an online survey will limit the amount of discussion possible.

panels[2 3] and participating in a James Lind Alliance Priority Setting Partnership for Sight Loss and Vision,[4] but we have largely relied on review author teams to suggest review titles. These titles are evaluated by the editorial base and our network of editors (see https://eyes.cochrane.org/about-cev for list) to assess whether they would form a suitable review question, ensuring no overlap with current Cochrane systematic reviews or with high quality, recently published, non-Cochrane systematic reviews. We consider the following criteria when prioritising review titles suggested by review author teams:

► Does the proposed new review (or review update) address an important clinical uncertainty? By 'important' we mean that the review topic is one that patients, clinicians, policymakers or the public would like to have answered, that is, is important to them. A 'clinical uncertainty' reflects the situation where there is evidence of variation in practice or differing opinions as to the best or most effective intervention.
► Will a Cochrane Review (new review or review update) at this point in time resolve this clinical uncertainty? Largely this means that we aim to prioritise reviews that will include a number of

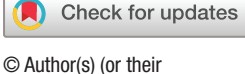

For numbered affiliations see end of article.

**Correspondence to**
Dr Jennifer R Evans;
jennifer.evans@qub.ac.uk

## INTRODUCTION

Cochrane Eyes and Vision (CEV) is an international network of individuals working to prepare, maintain and promote access to systematic reviews of interventions to treat or prevent eye diseases or vision impairment. We also conduct systematic reviews of the accuracy of diagnostic tests for common ocular diseases or conditions.

CEV was established in 1997.[1] We have used a variety of priority setting approaches over the years, including working with guideline



reasonably large and robust studies. However, sometimes we judge that identifying a gap in the evidence is also important, if identifying that gap will be likely to lead to further intervention studies.

▶ To what extent would resolving this clinical uncertainty reduce the magnitude of vision impairment and eye health disorders? This is clearly a subjective judgement but takes into account knowledge of the magnitude of the eye disease or vision impairment and anticipated effect, or cost-effectiveness, of the intervention.

▶ To what extent would resolving this clinical uncertainty reduce inequalities/disparities in the magnitude of disease or access to care for vision impairment or eye health disorders? Ideally, we would prioritise reviews and interventions that address inequity.

This approach identifies titles that are important to clinicians and researchers but may be less likely to identify questions relevant to other stakeholders such as patients, public and policy-makers. It is also likely to result in focus on higher-income settings, with high research capacity and less emphasis on equity-relevant titles. A recent review highlighted the lack of equity-relevant Cochrane Reviews on cataract .[5]

CEV plans to undertake a priority setting exercise to assess systematically the nature and extent of research questions relevant to our scope, and to formally incorporate input from a wide range of stakeholders to set priorities for new and updated reviews. This document outlines the protocol for a priority setting exercise to identify important review questions. It is informed by guidance prepared by Cochrane on setting review priorities[6] and by the REporting guideline for PRIority SEtting of health research framework for reporting priority setting of health research.[7] Equity will be a specific consideration in the priority setting process. As part of this process, we will also be informed by the work of other organisations undertaking priority setting activities in the field of Eyes and Vision, including the recent Grand Challenges in global eye health undertaken as part of the *Lancet Global Health* Commission on Global Eye Health[8] and the Package of Eye Care Interventions (PECI) being developed by WHO.[9 10]

## AIMS

The CEV Priority Setting Exercise aims to generate and publicise a list of priority topics, for both new and updated reviews, ensuring involvement of our main stakeholders.

## METHODS
### Context and scope

The scope of CEV is global and includes conditions that are common and have a high global disease burden, for example, cataract and dry eye disease, and conditions that are rare but have a high impact on quality of life and high individual cost such as eye cancer. The intended

> **Box 1  List of conditions**
>
> 1. Cataract.
> 2. Refractive error.
> 3. Diabetic retinopathy.
> 4. Glaucoma.
> 5. Macular degeneration.
> 6. Amblyopia.
> 7. Disorders of eye movement.
> 8. Infectious and inflammatory diseases.
> 9. Ocular trauma.
> 10. Ocular surface disorders.
> 11. Disorders of the eyelid and lacrimal system.
> 12. Eye cancer.

beneficiaries of our work are people making healthcare decisions for eye healthcare. CEV reviews primarily address questions relevant to clinical research (treatment, diagnosis, prognosis) but also potentially cover public health, health services and implementation research.

We plan to focus on the list set out in box 1 adapted from conditions prioritised by WHO as part of the development of the PECI. These conditions have been selected based on a combination of data on disease magnitude, healthcare use and expert opinion.

### Governance

To advise on the scope of the exercise, we set up a steering group including ophthalmologists, optometrists, orthoptists, ophthalmic nurses and relevant professional bodies, patient organisations, experienced clinical editors, systematic review methodologists and information specialists of CEV; it includes participants from high and low-income settings. Members of the team have been involved in previous priority setting exercises.

### Stakeholders and participants

Any person with an interest in healthcare decision-making relevant to eye healthcare is eligible to take part. This includes patients, caregivers, the general public, health professions, researchers, policy-makers, government and non-government organisations, and industry. We will actively seek out potential stakeholders using the expertise of the steering group to identify relevant stakeholder organisations and individuals globally in their field. We will then write to each stakeholder organisation or individual identified by the steering group to invite them to take part in the online survey and to identify further relevant stakeholders (snowballing).

### Identifying research questions

For each condition, we will undertake the following steps in order to identify priority questions for each condition, consulting with the steering group as needed (figure 1):
1. Identify and summarise relevant data on research in eyes and vision. We plan to use the following sources:
▶ Global policy reports.
▶ Other research prioritisation and roadmaps.

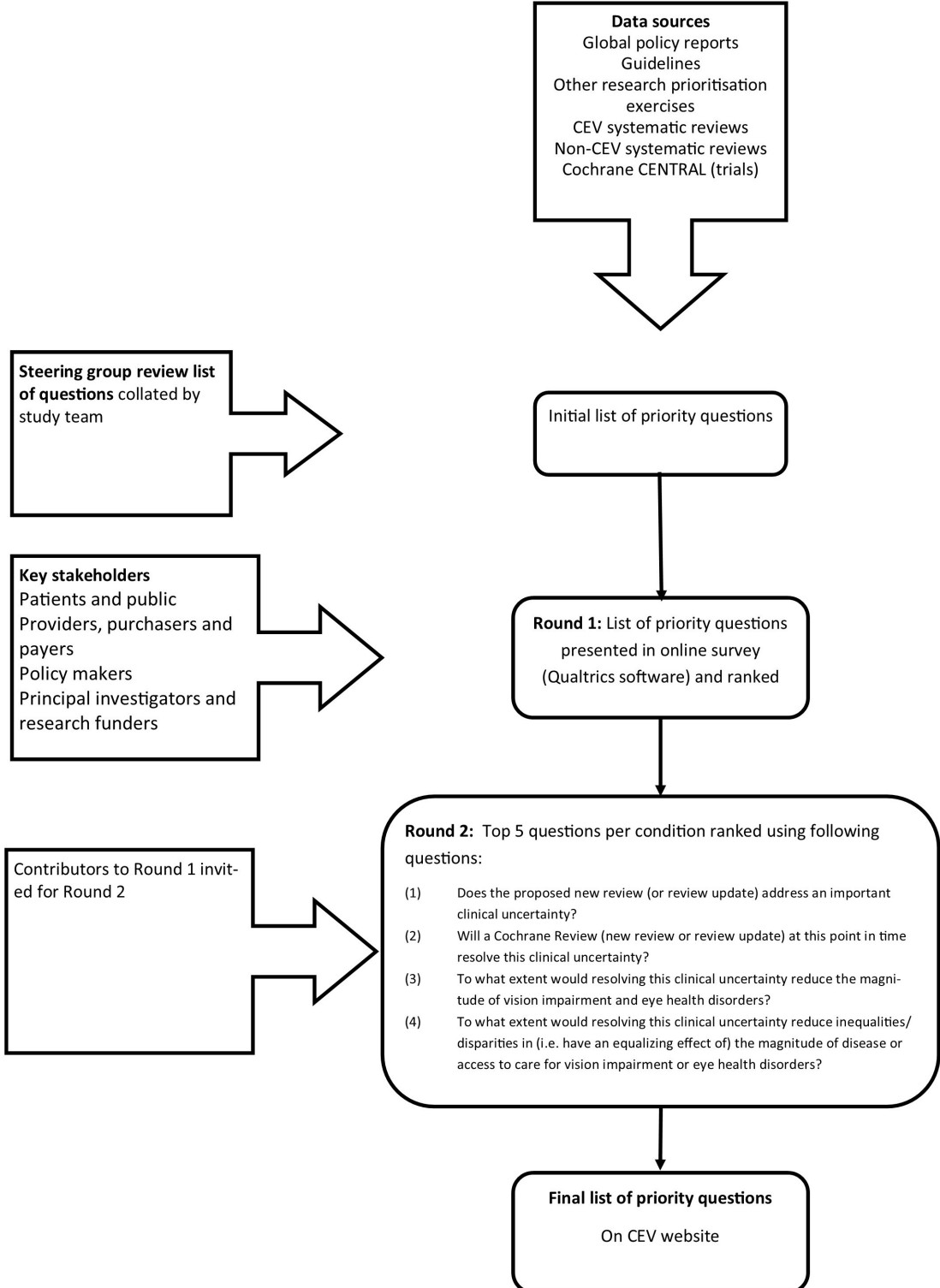

**Figure 1** Flow chart of study process. CEV, Cochrane Eyes and Vision.

► Guidelines.
► CEV systematic reviews.
► CEV@US Project Database of Systematic Reviews in Eyes and Vision.[11]
► Intervention studies on The Cochrane Library (CENTRAL).

We will identify global policy reports and other research prioritisation exercises (eg, James Lind Alliance Priority Setting Process in Sight Loss and Vision[4]) from searching academic databases and contact with our steering group. One member of the research team will scrutinise these reports for identified research questions. We will identify high-quality guidelines from the process followed by WHO during the development of the PECI. We will rank current CEV reviews according to impact using Altmetric and access statistics provided by the publisher. The top

20% of reviews identified by this process, that is, reviews with highest Altmetric score and/or access statistics will be eligible for the priority setting process. We will search The Cochrane Library for studies relevant to Eyes and Vision published within the last 3 years, or on a trials registers. One author (IG) will scan these reports to identify new potential review titles, that is, PICOs identified in two or more studies, that have not already been addressed in Cochrane Reviews. Search strategies are in the online supplemental appendix 1.

(2) Using the information from (1) we will prepare a list of potential review topics, removing duplicates as needed or merging/combining similar questions. This list will be reviewed by the Steering Group after it has been refined by removing questions where:

► High quality, current systematic reviews exist and new trials are either unlikely to have been done, or if they have been done, are unlikely to change the conclusions of the review. We will check for Cochrane Reviews and non-Cochrane high quality systematic reviews in the database maintained by CEV@US project.
► Topic is beyond scope because either it does not address a condition of interest (box 1) or it does not address a relevant clinical question.
► Question is unclear or ill defined or cannot be answered by either an intervention, diagnostic test accuracy, prognostic or scoping review.

### Conducting the priority setting exercise

We will seek the views of external and internal stakeholders on the list generated in (2) by conducting an online survey using Qualtrics software (Qualtrics, Provo, Utah, USA, available at www.qualtrics.com). We aim to present a list of 10–15 potential review questions (new and/or updates) for each condition group in this online survey.

We will identify key stakeholders, initially by consultation with the Steering Group, but also including further 'snowballing', that is, asking stakeholders to identify other relevant stakeholders. We will request information on key research needs. The categories of stakeholders that we will consider include but will not be restricted to:

► Patients and public.
► Providers, purchasers and payers.
► Policy-makers.
► Principal investigators and research funders.

Participants will be identified through two methods: advertising and direct invitation. We will advertise through social media (eg, Twitter, Facebook and Cochrane channels) and within academic (eg, Alumni networks, Community Eye Health Bulletin,) clinical and professional networks (eg, ophthalmological, optometric and orthoptic societies). We will inform the stakeholders identified above, and ask members of the Steering Group, to circulate within their networks. We will invite some participants directly to take part and will ensure that these invitations are balanced with respect to gender, location (working/living in a high-income or

lower-income setting) and profession (clinician, patient, policy-maker). We will approach Cochrane contributors and other contacts we consider potentially interested to contribute and will draw on the previous participants of the Grand Challenges for Eye Health (names available in public domain). We will not perform any other formal process for ensuring balance, but we will collect limited information on respondents (gender, location, profession) and how they were informed about the survey so that we understand who has responded.

We will conduct a two-round process. In the first round, we will present questions separately for each condition and ask the participants to rank in order of priority, that is, which reviews or review updates should CEV complete first, in the opinion of the respondent. There will be space for the participant to add additional questions that have not been included in the presented list. We will be seeking questions relevant to intervention reviews, diagnostic test accuracy reviews, prognostic reviews or scoping reviews only. Within 4 weeks, we will present a second round, in which the top five questions for each condition will be presented, along with any additional new questions identified during the course of the first round. Participants will be asked to score each review question presented according to the following criteria (4-point scale 1=definitely not/no extent, 2=possibly not/small extent, 3=possibly yes/moderate extent, 4=definitely yes/large extent):

► Does the proposed new review (or review update) address an important clinical uncertainty?
► Will a Cochrane Review (new review or review update) at this point in time resolve this clinical uncertainty?
► To what extent would resolving this clinical uncertainty reduce the magnitude of vision impairment and eye health disorders?
► To what extent would resolving this clinical uncertainty reduce inequalities/disparities in (ie, have an equalising effect of) the magnitude of disease or access to care for vision impairment or eye health disorders?

Following these two rounds, for each condition, we will identify the three questions with the highest average score. Each question will be structured in the PICO (Participants, Interventions, Comparator, Outcome) format, adapted for other contexts, for example, diagnosis, prognosis as necessary. As part of this process, we will be guided by the quantitative and qualitative results of these surveys but, to align with our commitment to widen the inclusion and equity-relevance of CEV reviews, we will include at least one question relevant to lower income settings. We will also report ranking of review question priorities by location and stakeholder background to assess the extent to which priorities within different groups differ.

### Equity

We will consider equity as part of this process, drawing on methods developed by the Campbell and Cochrane Equity Methods group (https://methods.cochrane.org/

equity/about-us). One member of the Steering Group has a special interest in equity. Our main approach will be to ensure that we have as wide a participation as possible (see above for details). We are taking the opportunity to draw on current partnerships with global initiatives to ensure priorities are informed by representation from low-income and middle-income countries. We will also prioritise questions that the survey participants have considered would reduce inequalities (last question above).

## Patient and public involvement

Patient and public involvement will be through the Steering Group and by contributing to the priority setting exercise.

## ETHICS AND DISSEMINATION

The study has been approved by the ethics committee of the London School of Hygiene & Tropical Medicine. Please see online supplemental appendix 2 for information to be given to participants in the online survey. All data collection will be electronic. We will disseminate the findings through Cochrane channels and prepare a summary of the work for publication in a peer-reviewed journal.

We will publish, through relevant Cochrane channels, our intention to conduct a priority setting process so that external and internal stakeholders may be involved.

We plan to:
- ► Document our plans for priority setting including stakeholder engagement, methods and criteria.
- ► Document the implementation of the priority-setting process on our website (including link to relevant network portal) and in an academic publication.
- ► Publish a list of priority topics on the CEV website.
- ► Develop a plan for how the priority topics will be delivered.
- ► Provide feedback to stakeholders involved, including notification when priority reviews are published.

## Currency/timeframe

We plan to complete the Priority Setting Exercise during June to December 2021 and repeat within 3–5 years.

## Evaluation and feedback

Written feedback will be given to all participants in the process who have supplied an email address, including a plain language account of the process and outcome of the process. All participants will be acknowledged in the final report (with permission).

We will evaluate the priority setting process by asking participants to complete a questionnaire collecting quantitative data and qualitative information on the following outcomes.

### Short-term evaluation
- ► Did the priority setting process meet Cochrane mandatory and desirable criteria for governance, stakeholder engagement, documentation and dissemination?
- ► Was the process complete within the prespecified time frame?
- ► Was the process completed without using excessive CEV staff time?
- ► Gather feedback from stakeholders via questionnaire
    - What did stakeholders like about the process?
    - What did stakeholders want to improve about the process?
- ► Gather feedback from CEV staff
    - What did CEV staff like about the process?
    - What did CEV staff want to improve about the process?

### Long-term evaluation
1. Were the resultant reviews produced in a timely manner?
2. Were the resultant reviews relevant/important? For example, did they have higher Altmetric/impact score?
3. Were the reviews used in guidelines or other policy documents?

### Other considerations
- ► Equity—how have the results improved equity? Have any of the reviews considered most relevant for equity in the process above been undertaken?
- ► Has there been an increase in authors from low-income and middle-income settings?

## CONCLUSION

A systematic and transparent approach to identifying review questions, informed by credible evidence, and reaching out to a broader group of people to assess priorities will help CEV establish which reviews need to be prioritised in the next 3–5 years.

**Author affiliations**
[1]Centre for Public Health, Queen's University Belfast, Belfast, UK
[2]International Centre for Eye Health, London School of Hygiene & Tropical Medicine, London, UK
[3]Centre for Applied Vision Research, School of Health Sciences, City, University of London, London, UK
[4]Cochrane MOSS Network, Oxford University Hospitals NHS Trust, Oxford, UK
[5]Department of Health Services Research, University of Liverpool, Liverpool, UK
[6]Department of Surgery and Translational Medicine, University of Florence, Firenze, Toscana, Italy
[7]Anschutz Medical Campus, University of Colorado, Denver, Colorado, USA
[8]School of Optometry and Vision Science, University of Auckland, Auckland, New Zealand

**Collaborators** Cochrane Eyes and Vision (CEV) Priority Setting Steering committee, Augusto Azuara-Blanco (CEV Editor, Queen's University), Michael Bowen (College of Optometrists), Roxanne Crosby-Nwaobi (CEV Editor, Moorfields Eye Hospital), Jennifer Evans (CEV Co-ordinating Editor, UK editorial base), Barny Foot (Royal College of Ophthalmologists), Stephen Gichuhi (CEV Editor, University of Nairobi), Renata Gomes (Blind Veterans), John Lawrenson (CEV Co-ordinating Editor, City University), Tianjing Li (CEV Co-ordinating Editor, CEV@US), Roses Parker (Cochrane Network Support Fellow), Jacqui Ramke (CEV Editor with special interest in Equity,

University of Auckland/LSHTM), Fiona Rowe (CEV Editor, Liverpool University), Anupa Shah (CEV Managing Editor, UK editorial base), Gianni Virgili (CEV Co-ordinating Editor, UK editorial base and DTA satellite), Richard Wormald (Emeritus Co-ordinating Editor, UK editorial base).

**Contributors** All authors contributed to the conception and design of the study through participation in the steering group. JE wrote the first draft of the paper. JL, RP, FJR, GV, TL and JR revised it critically for important intellectual content. IG prepared the search strategies. All authors approved the submitted paper. As this is a protocol, acquisition, analysis and interpretation of data do not apply.

**Funding** Cochrane Eyes and Vision is supported by the National Institute for Health Research (NIHR) Cochrane Infrastructure grant NIHR129461 until end March 2021 and Northern Ireland Health and Social Care Agency from 1 April 2021. JR's appointment at the University of Auckland is funded by the Buchanan Charitable Foundation, New Zealand.

**Competing interests** TL directs the Cochrane Eyes and Vision US Project, supported by grant UG1EY020522 from the National Eye Institute, National Institutes of Health.

**Patient consent for publication** Not required.

**Provenance and peer review** Not commissioned; externally peer reviewed.

**ORCID iDs**
Jennifer R Evans http://orcid.org/0000-0002-6137-2030
Fiona J Rowe http://orcid.org/0000-0001-9210-9131
Jacqueline Ramke http://orcid.org/0000-0002-5764-1306

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
