## [Reviewer comments · BMJ Open]

ARTICLE DETAILS

TITLE (PROVISIONAL)	Identifying priority review questions for Cochrane Eyes and Vision: protocol for a Priority Setting Exercise
AUTHORS	Evans, Jennifer; Gordon, Iris; Lawrenson, John; Parker, Roses; Rowe, Fiona; Virgili, Gianni; Li, Tianjing; Ramke, Jacqueline

VERSION 1 – REVIEW

REVIEWER	Prakash Paudel University of New South Wales
REVIEW RETURNED	10-Feb-2021

GENERAL COMMENTS	In general, this protocol for a priority setting exercise for identifying priority review questions for CEV is well documented. However, I feel more clarity and explanation is required in methods and analysis section. The study protocol is also about the steps/processes leading to conducting a priority setting exercise, so including steps/processes in a flow-diagram will help readers to navigate and understand this protocol clearly. If possible and applicable, I suggest for briefing the steps of priority exercise into three main stages of Delphi process – the exploratory stage, the distillation stage and the utilisation stage. Authors state that three review questions (the final output) for each of the conditions in Box1 will be collected systematically. They mentioned about four steps/ways in methods and analysis (page 2, lines 23-27). However, the description in main text (page 5 line 41 onward) states three steps. Here I give my comments and suggestions on methods and analysis (as outlined into four steps). Please revise accordingly and as appropriate for presenting a clear flow of the steps or processes. 1) by contact with key stakeholders – it is better to have inclusion and exclusion criteria for selecting stakeholders (particularly nontechnical). The process of selection of participants and required balance or equity based on demographic characteristics is important. It would be clearer if this is framed in the protocol. As stated in page 7, line 35, the priority of representation and/or the outcome (3 review questions for each condition) is primarily for low income settings. Does this mean low- and middle-income countries? Will there be any proportionate participations from all levels or will it be higher rate (what %) of representation from low- and middle-income countries? Please clarify about this with appropriate reasoning. Aims and objectives (page 4, line 27); Are these aims for this proposed protocol or just priority setting exercise? Isn't no 1 and no 3 talking about achieving same outcome? Who are you referring for with 'our main stakeholders' in no 2 aim? why is this so specific? And, is no. 2 really an objective? Page 5, line 47: In first step, identify key stakeholders – this section
--

	does not explain/reflect their role in identifying research questions. If they are only for participating in the two-round process for priority setting exercise, why is it listed as no. 1 step in section - identifying research questions? With the statements in the Introduction, it looks like the first round of priority exercise was already undertaken (which identified 40 top challenges in global eye health). This Delphi exercise may sufficiently provide the pertinent research questions which can lead authors to two-round process of priority setting exercise (attach the questionnaire as supplementary material if this is the first step). Doesn't this exercise should be outlined as a first step of this protocol for a priority setting exercise? To me, it seems to be the first phase of Delphi use. And, with further addition of steps 2 & 3 described in page 6, authors will get the list of 15 review questions for each condition, and then comes the other two-round process described in page7, line 6 onward)? Page 7, line 8: who do you refer saying 'the panel' here? All stakeholders mentioned in step 1 or specific group? Page7, line 12: when will the second round happen? Do you mean after first round you again contact the stakeholders and send them the list with top 5 questions + additional questions for each condition obtained in the first-round? What will be the time gap for first and second round survey? How many follow-up participants (or say response rate) is estimated to take part in the second round? What is 4-point scale? Is it 1,2,3,4 or very important to less important or anything else? Please clarify. Even after doing the second-round, there is a third round, in which the steering group or authors group will review top three questions, and make their judgment by interfering the outcome of two-round process and prioritise questions relevant to lower settings!!?? In this context (Page 7, line 46-48), do you mean authors will keep (i.e., prioritise) the questions in the first place to reduce inequalities? 2) by scrutiny of global policy reports... - Please briefly mention what is the process of conducting this (by conducting literature review – search specific internet domains, websites of all eye health related organization and so on?), please clarify. Who will do this review (specific authors or ...xx... members of the steering group?)? How the agreed review outcome will be derived if ratings or decisions of members are different? In a statement (page 2, line 24), what do you mean by 'other priority setting exercises'? As mentioned in the Introduction (page 4, line 13), is this task a part of collaborations with the WHO vision programme? How is this protocol linked to WHO programme? Page 6, line 14; How and why high-quality guidelines will be used, and for what purpose? Who will rank CEV reviews and by what measurement scale - score or rank (and based on what?? – such as magnitude of the problem, importance of topics, socio-economic impact of the condition)? Only top 20% reviews will be used for the priority setting exercise – given top 20% how many questions for each condition will get selected? 3) by ranking current CEV reviews – Is there any guideline to rank the reviews? Who will rank this – authors or a separate expert team or members of steering group? How will this information be integrated into the priority setting exercise? It looks this ranking process is merged with the second step (review policy reports) when describing in Page 5, line 58 onward). Please report consistently. 4) by identifying questions addressed in intervention studies (published and ongoing) in The Cochrane Library. – Authors do not explain who will identify and on what basis? Any guideline or criteria for this? This approach also looks like a part of second step.
--	--

	Authors stated (Page 2, lines 28-30) that they will seek views (what specific views about – ranking with number, importance, VQoL impact or something else?) of external and internal stakeholders on the list of 12 x 15 = 180 questions (that is, 15 questions for each condition) by conducting an online survey. But how can nontechnical participants such as patients and policy makers give their opinion or rank the list. Please clarify the process how this biasness or uncertainty will be addressed or minimised. If need to be included, nontechnical/ nonprofessional participants' views and ranks should be presented separately, as they are likely to dilute or bias the priority settings. Importantly, inclusion/exclusion criteria of stakeholders (particularly nontechnical/nonprofessional participants) participating in an online survey is crucial. Article summary in Page 2 states - The focus will be mainly on new intervention reviews and topics? But later (Page 3, lines 14-26) says both new review and review updates? Page 2, lines 11-36; It states that the stakeholders will prioritise the review titles based on four criteria questions. Are nontechnical stakeholders capable to do this? If they are included in the review outcome process and if it is likely to get most responses from the participants in higher-income settings, it may result to biased outcome (the list of 3 review titles for each condition). How will authors address or minimise these issues? I suggest keeping a question - how the participants get informed about the survey (direct invitation email, social medias, snowballing etc)? What approach would be taken to ensure the survey reach worldwide? What is the estimated response rate for this survey when sending emails to stakeholders? Will there be tracking, or record of % of participants enrolled by a snowballing technique? The online survey itself also limits the response rate (possibly 30-50%). What mechanism will be placed to increase the response rate? Importantly, how do you reach to the stakeholders functioning at different level of settings? The experience gained by authors from the Delphi exercise (to identify grand challenges in global eye health) might help to come up with potential solutions to address low response rate in next two-round process. In ethics section in page 7, please mention how will you obtain participant's consent to enroll them in the online survey? Please include whether invitation letter with participant information sheet will be sent along with the survey link. Also, mention if paper record forms will be used in some situations. Please include the data analysis plan and mention what statistical analysis will be performed if there is plan for looking association with participants' demography. Please update the timeframe listed for priority setting exercise. How will authors do the short-term and long-term evaluation? How will you evaluate the equity and sustainability issues? A brief process on this exercise may be useful to include here. How will the feedback from the stakeholders gathered? Will authors send the evaluation report and ask them for their feedback?
--	--

REVIEWER	Ariela Gordon-Shaag Hadassah Academic College
REVIEW RETURNED	14-Apr-2021

GENERAL COMMENTS	This is an important endeavor and I congratulate you on aiming to make the review topics more equitable and addressing global concerns. I noted that you are collaborating with the WHO but did reference their Vision 2020 report:
---

	https://www.who.int/publications/i/item/9789241516570. The WHO states in this report that unaddressed refractive error is the most prevalent cause of avoidable vision impairment. This particularly impacts low- and middle-income countries. Yet you do not include refractive error in the list of priorities in Table 1. Thus, new diagnostic techniques or low-cost methods of spectacle correction would not be included in your review topics. I urge you to add refractive error to the list of priorities when you repeat the process in 3 and 5 years.
--	---

VERSION 1 – AUTHOR RESPONSE

Reviewer: 1

Dr. Prakash Paudel, University of New South Wales

Comments to the Author:

In general, this protocol for a priority setting exercise for identifying priority review questions for CEV is well documented. However, I feel more clarity and explanation is required in methods and analysis section.

Thank you for reviewing our paper and for your helpful comments. Please see our responses below.

The study protocol is also about the steps/processes leading to conducting a priority setting exercise, so including steps/processes in a flow-diagram will help readers to navigate and understand this protocol clearly. If possible and applicable, I suggest for briefing the steps of priority exercise into three main stages of Delphi process – the exploratory stage, the distillation stage and the utilisation stage.

We have added in a flow diagram (figure 1). As this priority setting exercise does not use the standard 3 step process of a classic Delphi method, we have removed reference to Delphi in describing the priority setting exercise.

Authors state that three review questions (the final output) for each of the conditions in Box1 will be collected systematically. They mentioned about four steps/ways in methods and analysis (page 2, lines 23-27). However, the description in main text (page 5 line 41 onward) states three steps.

Thank you for highlighting this discrepancy. We have edited the abstract (page 2, lines 23-27) to make more consistent with the structure of the main text.

Here I give my comments and suggestions on methods and analysis (as outlined into four steps). Please revise accordingly and as appropriate for presenting a clear flow of the steps or processes.

1) by contact with key stakeholders – it is better to have inclusion and exclusion criteria for selecting stakeholders (particularly nontechnical). The process of selection of participants and required balance or equity based on demographic characteristics is important. It would be clearer if this is framed in the protocol. As stated in page 7, line 35, the priority of representation and/or the outcome (3 review questions for each condition) is primarily for low income settings. Does this mean low- and middle-income countries? Will there be any proportionate participations from all levels or will it be higher rate (what %) of representation from low- and middle-income countries? Please clarify about this with appropriate reasoning.

We are reluctant to specify inclusion and exclusion criteria for key stakeholders because we want to be as inclusive as possible. We will reach out to relevant stakeholders, wherever they are based, but

also collect information to establish the extent to which we have been successful in engaging marginalised groups and people living and working in low and middle-income settings.

Aims and objectives (page 4, line 27); Are these aims for this proposed protocol or just priority setting exercise? Isn't no 1 and no 3 talking about achieving same outcome? Who are you referring for with 'our main stakeholders' in no 2 aim? why is this so specific? And, is no. 2 really an objective?

We have revised the aims and objectives as follows. "*The CEV Priority Setting Exercise aims to generate and publicise a list of priority topics, for both new and updated reviews, ensuring involvement of our main stakeholders in the process.*"

Page 5, line 47: In first step, identify key stakeholders – this section does not explain/reflect their role in identifying research questions. If they are only for participating in the two-round process for priority setting exercise, why is it listed as no. 1 step in section - identifying research questions?

We agree this is confusing. We have moved the paragraph on identifying stakeholders to the section on "conducting the priority setting exercise"

With the statements in the Introduction, it looks like the first round of priority exercise was already undertaken (which identified 40 top challenges in global eye health). This Delphi exercise may sufficiently provide the pertinent research questions which can lead authors to two-round process of priority setting exercise (attach the questionnaire as supplementary material if this is the first step). Doesn't this exercise should be outlined as a first step of this protocol for a priority setting exercise? To me, it seems to be the first phase of Delphi use. And, with further addition of steps 2 & 3 described in page 6, authors will get the list of 15 review questions for each condition, and then comes the other two-round process described in page7, line 6 onward)?

The Grand Challenges in Global Eye Health identified challenges for global eye health rather than questions for systematic reviews. In the context of identifying questions for systematic reviews, it is a potential source of review topics, alongside other sources, rather than a formal part of the current priority setting exercise for Cochrane Eyes and Vision. We have edited the information on the Grand Challenges in the introduction in order to avoid confusion.

Page 7, line 8: who do you refer saying 'the panel' here? All stakeholders mentioned in step 1 or specific group?

We have changed this to "participants" for consistency with the other sections.

Page7, line 12: when will the second round happen? Do you mean after first round you again contact the stakeholders and send them the list with top 5 questions + additional questions for each condition obtained in the first-round? What will be the time gap for first and second round survey? How many follow-up participants (or say response rate) is estimated to take part in the second round? What is 4-point scale? Is it 1,2,3,4 or very important to less important or anything else? Please clarify.

We have added information on when the second round will happen (within 4 weeks of the first round) and the 4-point scale (1=definitely not, 2=possibly not, 3=possibly yes, 4=definitely yes). It is difficult to estimate response rates but just to note that In the Grand Challenges overall 84% of people invited completed 3 rounds.

Even after doing the second-round, there is a third round, in which the steering group or authors group will review top three questions, and make their judgment by interfering the outcome of two-round process and prioritise questions relevant to lower settings!!?? In this context (Page 7, line 46-48), do you mean authors will keep (i.e., prioritise) the questions in the first place to reduce inequalities?

We have clarified this. We now state that we will select at least one question relevant to a LMIC setting for each condition.

2) by scrutiny of global policy reports... - Please briefly mention what is the process of conducting this (by conducting literature review – search specific internet domains, websites of all eye health related organization and so on?), please clarify. Who will do this review (specific authors or ...xx... members of the steering group)? How the agreed review outcome will be derived if ratings or decisions of members are different? In a statement (page 2, line 24), what do you mean by 'other priority setting exercises'? As mentioned in the Introduction (page 4, line 13), is this task a part of collaborations with the WHO vision programme? How is this protocol linked to WHO programme?

We have added in the search strategies in an appendix and clarified who will do the searching. We have added in an example of another priority setting process. This protocol is using a similar approach as the WHO programme to specifying conditions and identifying guidelines but is not otherwise formally linked. To make this clearer, we have edited the introduction removing reference to "with whom we are working" which refers to work outside the remit of this specific exercise.

Page 6, line 14; How and why high-quality guidelines will be used, and for what purpose? Who will rank CEV reviews and by what measurement scale - score or rank (and based on what?? – such as magnitude of the problem, importance of topics, socio-economic impact of the condition)? Only top 20% reviews will be used for the priority setting exercise – given top 20% how many questions for each condition will get selected?

As for other background reports, we will use the guidelines for the purpose of identifying potential clinical uncertainties identified by the guideline panels that might lead to impactful review questions. Similarly, the most impactful CEV reviews as judged by numeric ranking of Altmetric score and access statistics, will be used as a source of questions. We aim to include approximately 15 questions per condition.

3) by ranking current CEV reviews – Is there any guideline to rank the reviews? Who will rank this – authors or a separate expert team or members of steering group? How will this information be integrated into the priority setting exercise? It looks this ranking process is merged with the second step (review policy reports) when describing in Page 5, line 58 onward). Please report consistently.

We have clarified that current CEV reviews will be ranked numerically in order of Almetric score and access statistics. This does not need to be done by an expert team as it is simply a spreadsheet exercise. The top 20% of review questions identified will be added to the other review questions and managed as outlined in section (2).

4) by identifying questions addressed in intervention studies (published and ongoing) in The Cochrane Library. – Authors do not explain who will identify and on what basis? Any guideline or criteria for this? This approach also looks like a part of second step.

This is part of (and outlined in) the first step of identifying potential review questions. The second step is to curate a final list. New review questions with 2 or more relevant studies that have not already been included in Cochrane reviews will be selected.

Authors stated (Page 2, lines 28-30) that they will seek views (what specific views about – ranking with number, importance, VQoL impact or something else?) of external and internal stakeholders on the list of $12 \times 15 = 180$ questions (that is, 15 questions for each condition) by conducting an online survey. But how can nontechnical participants such as patients and policy makers give their opinion or rank the list. Please clarify the process how this biasness or uncertainty will be addressed or minimised. If need to be included, nontechnical/ nonprofessional participants' views and ranks should be presented separately, as they are likely to dilute or bias the priority settings. Importantly,

inclusion/exclusion criteria of stakeholders (particularly nontechnical/nonprofessional participants) participating in an online survey is crucial.

The online survey will ask respondents to rank the questions in order of importance using a 'drag and drop' question format. Each condition will be ranked separately. We are interested in the views of people with a lived experience of the conditions – in Cochrane the term used is 'consumers'. In other priority setting exercises in Sight Loss and Vision, for example James Lind Alliance, views of consumers have been included successfully. In the current survey, we agree that priorities identified by consumers may be different to priorities of researchers or health care providers and, if so, this will be an important finding of the priority setting exercise.

Article summary in Page 2 states - The focus will be mainly on new intervention reviews and topics? But later (Page 3, lines 14-26) says both new review and review updates?

Thank you for highlighting this discrepancy. We have edited the Article summary.

Page 2, lines 11-36; It states that the stakeholders will prioritise the review titles based on four criteria questions. Are nontechnical stakeholders capable to do this? If they are included in the review outcome process and if it is likely to get most responses from the participants in higher-income settings, it may result to biased outcome (the list of 3 review titles for each condition). How will authors address or minimise these issues?

Thank you for this comment. It is true that contributors to the priority setting process will have a variety of backgrounds. We aim to be inclusive and involve patients and the public (consumers) in the process. As part of the survey, we will collect information on the characteristics of the respondents and will be able to determine the extent to which selected priorities are related to background of respondent.

I suggest keeping a question - how the participants get informed about the survey (direct invitation email, social medias, snowballing etc)?

Thank you for this suggestion. We have included the following text "*we will collect limited information on respondents (gender, location, profession) and how they were informed about the survey*"

What approach would be taken to ensure the survey reach worldwide?

Our main approach will be to use our academic, clinical and patient group networks which have a wide global reach, and by asking contributors to circulate within their networks. We will also approach relevant patient organisations and advertise on social media. See page 6 last paragraph.

What is the estimated response rate for this survey when sending emails to stakeholders?

Unfortunately, we do not know in advance what the response rate will be. Experience with the Grand Challenges suggests that 84% of invited participants completed all 3 rounds of a Delphi process.

Will there be tracking, or record of % of participants enrolled by a snowballing technique?

We have included a question on this (see above) so we should be able to identify this percentage.

The online survey itself also limits the response rate (possibly 30-50%). What mechanism will be placed to increase the response rate? Importantly, how do you reach to the stakeholders functioning at different level of settings? The experience gained by authors from the Delphi exercise (to identify grand challenges in global eye health) might help to come up with potential solutions to address low response rate in next two-round process.

We accept the limitations of an online survey but do not have the resources to use face to face methods at the moment.

In ethics section in page 7, please mention how will you obtain participant's consent to enroll them in the online survey? Please include whether invitation letter with participant information sheet will be sent along with the survey link. Also, mention if paper record forms will be used in some situations.

The consent is part of the online form. We have added the information to be provided to participants in an appendix. We have clarified that all information is to be collected electronically.

Please include the data analysis plan and mention what statistical analysis will be performed if there is plan for looking association with participants' demography.

We do not plan formal statistical analyses but we will report ranking of review question priorities by location and stakeholder background to assess the extent to which priorities within different groups differ. We have amended the text to make this clearer.

Please update the timeframe listed for priority setting exercise.

We have updated the timeframe to June to December 2021.

How will authors do the short-term and long-term evaluation? How will you evaluate the equity and sustainability issues? A brief process on this exercise may be useful to include here.

How will the feedback from the stakeholders gathered? Will authors send the evaluation report and ask them for their feedback?

We will ask the stakeholders to complete a questionnaire. We have clarified this.

Reviewer: 2

Dr. Ariela Gordon-Shaag, Hadassah Academic College

Comments to the Author:

This is an important endeavor and I congratulate you on aiming to make the review topics more equitable and addressing global concerns. I noted that you are collaborating with the WHO but did reference their Vision 2020 report: <https://www.who.int/publications/i/item/9789241516570>. The WHO states in this report that unaddressed refractive error is the most prevalent cause of avoidable vision impairment. This particularly impacts low- and middle-income countries. Yet you do not include refractive error in the list of priorities in Table 1. Thus, new diagnostic techniques or low-cost methods of spectacle correction would not be included in your review topics. I urge you to add refractive error to the list of priorities when you repeat the process in 3 and 5 years.

Thank you for reviewing our paper and for your helpful comments. We agree that refractive error is an important cause of avoidable visual impairment globally and should be included in this exercise. The conditions in box 2 are listed in order of global burden and refractive error thus appears as condition number 2 (after cataract). We have added in the reference to the World Report on Vision.

VERSION 2 – REVIEW

REVIEWER	Prakash Paudel University of New South Wales
REVIEW RETURNED	17-May-2021

GENERAL COMMENTS	The revised protocol is clearly presented, the process is logical and simple to understand. Inclusion of figure summarise the study protocol: aim, methods, and expected outcome. A few potential issues: patient and public involvement process is with the steering committee decision, so it is mystery and not explained; and a little more information/clarity needed for second round process. With these few minor revisions/ clarifications, I think this protocol is all good. Page 24 of 28, line 29-30; The participants will be asked to rank the questions in order of priority for (or importance of) what??. Page 24 of 28, line 37-50; the first two questions can have the described 4-point scale (definitely/possibly). However, to what extent questions, the scale would be different. Example: high, some-what, low and not at all. Please clarify/ revise. Page 24 of 28, line 52-54; How is the highest average score calculated? If 'definitely Yes' for first two questions and responded 'to high extent' for next 2 questions, this gives the highest possible score (4+4+4+4=16). I understand this will be the score calculation. However, there is one thing which is still not clear from the protocol. I hope some clarity comes from the authors. All together 60 questions (12 conditions with 5 questions for each) will be listed in second survey and they must be scored using the scale range 1-4 by responding 4 questions. Are these four questions used for all 60 questions separately or for 12 conditions individually (i.e., for set of 5 questions overall)? The wordings of four criteria questions can be same (as of now), if these are for all 12 conditions (with set of 5 questions). But the language of the four questions needs to be changed for each, if these are presented for 60 questions separately (as final output is top 3 questions for each condition). Page 16 of 28, line 22-23: is this only for topics relevant to refractive error?? or all 12 conditions? I believe it should be for all 12 conditions. Please clarify/revise. Page 16 of 28, line 29-30: I did not understand here, which separate surveys are available? Isn't this information sheet for this online survey which includes questions for all 12 conditions and ask them to rank the questions for all conditions?
---

VERSION 2 – AUTHOR RESPONSE

Reviewer: 1

Dr. Prakash Paudel, University of New South Wales

Comments to the Author:

The revised protocol is clearly presented, the process is logical and simple to understand. Inclusion of figure summarise the study protocol: aim, methods, and expected outcome. A few potential issues: patient and public involvement process is with the steering committee decision, so it is mystery and not explained; and a little more information/clarity needed for second round process. With these few minor revisions/ clarifications, I think this protocol is all good.

Thank you for your time reviewing our paper and for your very helpful comments.

Page 24 of 28, line 29-30; The participants will be asked to rank the questions in order of priority for (or importance of) what??.

We are seeking opinions from our stakeholders as to which would be the most useful or important reviews to do first. For the respondents, what is important will vary according to their background and

context so we do not seek to define importance for them. We have clarified this in the paper by adding the following text. “.. that is, which reviews or review updates should Cochrane Eyes and Vision complete first, in the opinion of the respondent.”

Page 24 of 28, line 37-50; the first two questions can have the described 4-point scale (definitely/possibly). However, to what extent questions, the scale would be different. Example: high, some-what, low and not at all. Please clarify/ revise.

We have added extra wording to the scale as follows: 4-point scale 1=definitely not/no extent, 2=possibly not/small extent, 3=possibly yes/moderate extent, 4=definitely yes/large extent

Page 24 of 28, line 52-54; How is the highest average score calculated? If ‘definitely Yes’ for first two questions and responded ‘to high extent’ for next 2 questions, this gives the highest possible score (4+4+4+4=16). I understand this will be the score calculation.

Yes that is correct.

However, there is one thing which is still not clear from the protocol. I hope some clarity comes from the authors. All together 60 questions (12 conditions with 5 questions for each) will be listed in second survey and they must be scored using the scale range 1-4 by responding 4 questions. Are these four questions used for all 60 questions separately or for 12 conditions individually (i.e., for set of 5 questions overall)? The wordings of four criteria questions can be same (as of now), if these are for all 12 conditions (with set of 5 questions). But the language of the four questions needs to be changed for each, if these are presented for 60 questions separately (as final output is top 3 questions for each condition).

We have clarified in the paper that each proposed review question will be scored.

Page 16 of 28, line 22-23: is this only for topics relevant to refractive error?? or all 12 conditions? I believe it should be for all 12 conditions. Please clarify/revise.

Thank you. We have modified. We will present these in groups to make the survey easier to manage.

Page 16 of 28, line 29-30: I did not understand here, which separate surveys are available? Isn't this information sheet for this online survey which includes questions for all 12 conditions and ask them to rank the questions for all conditions?

We have clarified by making the following modification to this section of the appendix.

What is involved in taking part in the study? We would like you to take part in two rounds by online questionnaire. This is the first round and we expect it will take around 10 minutes to do. On the following pages, you will be presented with a list of potential review topics relevant to refractive error and asked to rank them in order of importance. These potential review topics were identified by systematic searching of global policy reports, guidelines and reports of relevant reviews and studies. We may have missed important questions and so there will also be an opportunity to tell us of priority topics that are not on the list.

Different eye conditions are considered separately and there are separate surveys available for other eye conditions (cataract, glaucoma etc).